# Data-driven predictions of potential Leishmania vectors in the Americas

**Gowri M. Vadmal**[1]*, **Caroline K. Glidden**[1], **Barbara A. Han**[2], **Bruno M. Carvalho**[3], **Adrian A. Castellanos**[2], **Erin A. Mordecai**[1]

**1** Department of Biology, Stanford University, Stanford, California, United States of America, **2** Cary Institute of Ecosystem Studies, Millbrook, New York, United States of America, **3** Climate and Health Program, Barcelona Institute for Global Health, Barcelona, Spain

* gvadmal@stanford.edu

**Data Availability Statement:** All data and code files are available from the github repository (https://github.com/mudkins/ACL-vector-data-analysis).

## Abstract

The incidence of vector-borne diseases is rising as deforestation, climate change, and globalization bring humans in contact with arthropods that can transmit pathogens. In particular, incidence of American Cutaneous Leishmaniasis (ACL), a disease caused by parasites transmitted by sandflies, is increasing as previously intact habitats are cleared for agriculture and urban areas, potentially bringing people into contact with vectors and reservoir hosts. Previous evidence has identified dozens of sandfly species that have been infected with and/or transmit *Leishmania* parasites. However, there is an incomplete understanding of which sandfly species transmit the parasite, complicating efforts to limit disease spread. Here, we apply machine learning models (boosted regression trees) to leverage biological and geographical traits of known sandfly vectors to predict potential vectors. Additionally, we generate trait profiles of confirmed vectors and identify important factors in transmission. Our model performed well with an average out of sample accuracy of 86%. The models predict that synanthropic sandflies living in areas with greater canopy height, less human modification, and within an optimal range of rainfall are more likely to be *Leishmania* vectors. We also observed that generalist sandflies that are able to inhabit many different ecoregions are more likely to transmit the parasites. Our results suggest that *Psychodopygus amazonensis* and *Nyssomia antunesi* are unidentified potential vectors, and should be the focus of sampling and research efforts. Overall, we found that our machine learning approach provides valuable information for *Leishmania* surveillance and management in an otherwise complex and data sparse system.

## Author summary

American Cutaneous Leishmaniasis (ACL) is a neglected disease caused by sandfly-transmitted parasites in the Americas. There is an incomplete understanding of which sandfly species transmit the parasite, complicating efforts to limit parasite transmission and consequently, disease burden. In this study, the authors created a database of sandfly traits, then used predictive models to determine important factors in parasite transmission and how different climate and environmental variables predict which vectors can transmit the

This link is also within the Supporting Materials section.

**Funding:** This project was supported by the Stanford King Center for Global Development (GMV, EAM), the National Science Foundation (DEB-2011147 with the Fogarty International Center, CKG, EAM; DEB-1717282, BAH, AAC), the National Institutes of Health (R35GM133439, R01AI168097, and R01AI102918 EAM; 5U01AI15180703, BAH, AAC), the Stanford Woods Institute for the Environment, the Stanford Center for Innovation in Global Health, the Severo Ochoa Center of Excellence Grant, Spanish Ministry of Science and Innovation & Spanish State Research Agency (CEX2018- 000806-S, BMC). EAM was additionally supported by seed grants from the Stanford Woods Institute for the Environment, King Center on Global Development, Center for Innovation in Global Health, and the Terman Award. The funders had no role in study design, data collection and analysis, decision to publish, or preparation of the manuscript.

**Competing interests:** The authors declare that they have no conflict of interest.

parasites that cause ACL. The models suggest that transmission occurs at the interface between domestic habitats and well-preserved forests. The authors also generate predictions of which sandflies might be transmitting the parasites that are not known vectors at the time, specifically *Psychodopygus amazonensis* and *Nyssomia antunesi*. This new knowledge can lead to a better understanding of the system of transmission and can point to possible hotspots of risk. The analysis can also help direct researchers to areas of interest for sampling studies, as well as specific sandflies on which to focus effort.

## Introduction

American cutaneous leishmaniasis (ACL) is a neglected tropical disease caused by parasites in the genus *Leishmania* and transmitted by sandflies of the subfamily Phlebotominae [1,2]. The World Health Organization estimates that worldwide there are approximately 1 to 2 million new cases of leishmaniasis each year [3], with 700,000 to 1 million of those cases identified as cutaneous leishmaniasis cases [4]. ACL cases occur across the Americas, with hotspots in northeastern and southeastern states in Mexico, northern Nicaragua, Costa Rica, Brazil, Peru, and at the convergence of the borders of Brazil, Peru, and Bolivia [5,6]. In some regions, incidence of ACL is increasing among farmers, loggers, hunters, and others working at the forest-human interface. Additionally, although primarily a tropical and subtropical disease [2], cases have also been more recently reported in the southern United States [3,7], making it an important emerging health problem in temperate regions.

Similar to other disease-causing parasites, *Leishmania* parasites are found in wild and domestic reservoir hosts, which are located across the Americas, and are picked up by sandfly females during their blood meal before laying eggs [2,8,9]. In the female sandfly gut, the parasite amastigotes develop into promastigotes, which migrate to the salivary glands and spread to other mammals or to humans during subsequent blood meals [2,8,9] (Fig 1). Once a human is

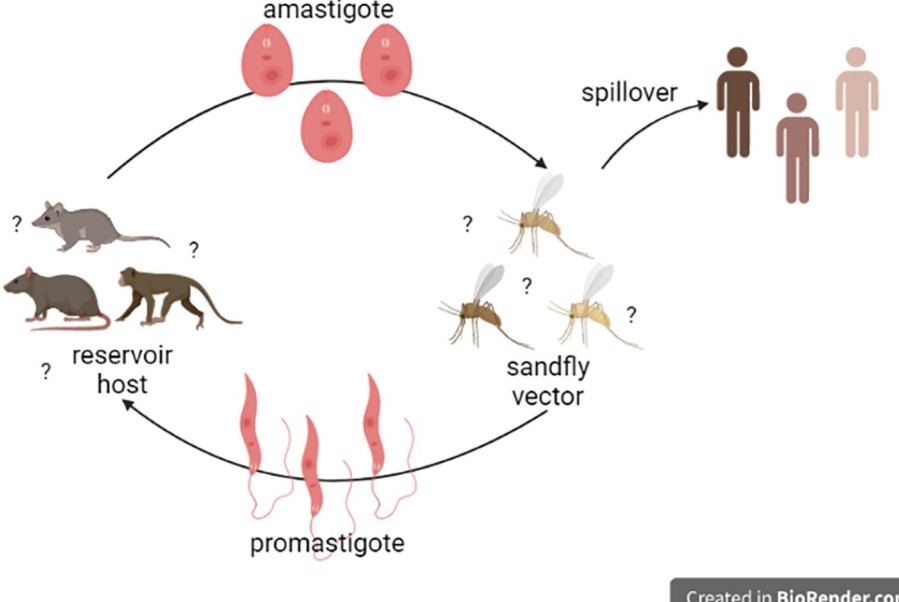

**Fig 1. ACL life cycle.** Female sandfly vectors pick up *Leishmania* parasites during their blood meal from reservoir hosts. Spillover events occur when a sandfly with *Leishmania* parasites in its salivary glands takes a blood meal from a human and infects the human with the parasite. Identity of all reservoir hosts and sandfly vectors are still unknown, which makes it hard to model and prevent transmission. Created with BioRender.com.

infected, the incubation period typically lasts around one to ten weeks (but can last many years) in which promastigotes invade local tissues and transform into amastigotes, entering macrophages through phagocytosis [2,8]. Clinical symptoms include lesions, rashes, open sores, ulcers, and small bumps covering the skin, which can lead to deformation with possible recurrences [2]. In some cases, ACL can evolve into diffuse or disseminated leishmaniasis, and rarely, into mucocutaneous leishmaniasis which can lead to severe facial mutilations and extensive disfiguring of the face, soft palate, pharynx, and larynx [2,10,11]. Overall, the disease is extremely painful and at times, severely debilitating.

There is no recognized oral treatment of ACL, and current antimonial treatments are painful, potentially dangerous and expensive [2,3,8,11,12]. Therefore, ACL is best managed through ecological interventions like controlling vectors and preventing transmission [13].

Climate change, deforestation, travel, and natural disasters are correlated with the spread of the parasite and leishmaniasis [2], yet it is hard to mechanistically predict the effect of global change on leishmaniasis incidence and distribution due to gaps in understanding of the parasite transmission cycle. Specifically, the full suite of reservoir hosts and confirmed vectors has not been fully characterized, making it difficult to model transmission under global change. For example, the incubation period of the parasite in the vector can be longer at lower temperatures and *shorter* at higher temperatures, ultimately impacting the number of hosts one sandfly can infect [2,14,15]. However, temperature *responses* likely vary by sandfly species [14–16], and thus the effect of climate warming on *Leishmania* transmission cannot be predicted precisely as the full range of sandflies that could transmit the parasites have yet to be described. Additionally, since different sandfly species have different habitat and biting preferences, sandfly species differ in their propensity to respond to land use change and contact infectious reservoir hosts and humans, thereby contributing to the human ACL burden. These are key uncertainties that need to be resolved for a more complete understanding of the transmission cycle, which will help to inform these mechanisms and identify vector and disease management opportunities.

Parasite-vector interactions can be divided into restrictive (sandflies that demonstrate specificity for the *Leishmania* species they can transmit) and permissive (sandflies that show non-specific interactions) groupings under laboratory conditions [17]. While restrictive pairs are well studied and linked together using molecular analysis methods, not much is known about permissive interactions, and they may be under-represented in the known list of sandfly vectors [17]. Thus, the current classification criteria might overlook some sandfly vectors that carry several different species of *Leishmania*. There have been efforts to use modeling approaches to identify environmental factors and vectors at the local level in Colombia [18,19] and in the Middle East [20], but no such models have been built using data from across the Americas.

In recent years, new machine learning methods have been used to identify potential reservoir hosts, vectors, and important factors in transmission of other vector-borne pathogens [21,22]. In order to fill in gaps in our understanding of *Leishmania* transmission, we use a similar approach to model the relationships between sandfly biology and vector status. From these models, we generate a list of sandflies predicted to transmit *Leishmania* spp. causing ACL, which we recommend should be empirically tested for vector competence and added to surveillance efforts.

## Methods

### Data collection

We first compiled a database on Phlebotomine sandfly vector status as well as behavioral, morphological, taxonomic, and ecological traits that could be used to delineate vectors from non-

vectors. Past vector identification has typically relied on the following criteria stated by Killick-Kendrick: (i) epidemiological observation that the sandfly is anthropophilic, (ii) proof that the fly feeds regularly on a relevant reservoir host, (iii) repeated isolation and identification of the same species of *Leishmania* spp. promastigotes infecting the humans in the surrounding area, (iv) evidence that the fly supports the complete development of the parasite, (v) and experimental evidence the sandfly can transmit the parasite through blood meal bite [17,23,24]. We defined a known vector to be a sandfly that is incriminated as a *Leishmania* vector by these criteria, which are generally considered to be the gold standard for identification [23]. We directly used vector status from Akhoundi et al. [24], which used the above criteria for identification. Molecular methods have been discussed as possible evidence, but are not sufficient to incriminate a sandfly species as a proven vector [24]. Here, we will use the term 'vector' to represent the sandflies that are confirmed through the five criteria stated by Killick-Kendrick [23]. We note that our definition of vector is specific to vectors of *Leishmania* spp. that can infect humans. We defined a 'potential vector' to be a sandfly that has been observed to carry *Leishmania* parasites in the wild via molecular diagnosis or dissection but not confirmed to transmit it to humans; it is important to note that we did not consider sandflies that have been experimentally infected with a *Leishmania* parasite to be a potential vector, and that we did not apply positive labels to potential vectors in our primary analysis. Our dataset comprised 512 documented sandfly species across the Americas (sample size n = 512). Thirty-seven of these 512 are confirmed at the time of analysis as vectors of one or more species of *Leishmania* causing ACL, and 35 additional species are potential vectors [10,24–28].

From the literature we collected thirteen female morphological traits [29–32] including but not limited to wing length, width, and number of teeth (S1 Table), as well as vector and infection status [22,26,27,30,33–37]. We included genus as a variable to include a measure of taxonomic relatedness in the model. Most of the sandfly morphological traits and biting behavior were taken from Young & Duncan's 1994 book [30], with additional biting behavior data taken from sandfly surveys using Disney and Shannon traps [25,27,37–77]. We paired occurrence points from the Global Biodiversity Information Facility (GBIF) [78,79] and published sandfly sampling studies [37–77] with GIS data from Google Earth Engine to describe biogeographical features of sandfly habitat such as temperature, wind speed, and canopy height (S1 Table). For each sampling study from the literature, we only included sandflies that made up more than 1% of the sampled species to account for possible misidentification or outliers. Habitat features were calculated for a sandfly species if there were at least 4 occurrence points for the species.

We used the Google Earth Engine Python API in Jupyter Notebook to get environmental and geographical traits averaged over each species' distribution, using the smallest spatial resolution possible as sandfly dispersal is very limited [2,80,81] (S1 Table). Based on known sandfly-parasite interactions, we expect important traits to include temperature, forest integrity, and other environmental change variables [2,4,14–16]. We used Copernicus Climate Change Service's ERA5 datasets of biogeographical data to get the mean monthly temperature, temperature range, mean monthly total rainfall, mean monthly wind speed, and mean elevation [82]. All data from Google Earth Engine datasets was from 2009 to 2019. We used NASA's Terra Vegetation dataset to get an enhanced vegetation index [83] and the Copernicus Global Land Cover dataset for tree, shrub, urban, grass, and water cover [84]. We used NOAA's ETOPO1 dataset for elevation [85] and NASA and JPL's Global Forest Canopy Height for canopy height [86]. We defined a species' ecoregion breadth as the number of different ecoregions it inhabited in the RESOLVE Biodiversity and Wildlife Solutions dataset (i.e., how many unique ecoregions the occurrence points mapped to) [87]. A species' presence in a biome was a binary trait for 10 different biomes [87]. The global human modification trait was the cumulative measure

of human modification of terrestrial lands globally at a 1 square-kilometer resolution in the Conservation Science Partners gHM dataset [88]. We used the Forest Landscape Integrity Index for the average forest integrity of an occurrence point, determined by degree of anthropogenic modification [89]. We used temperature variance as an indicator of seasonality, and found the average variance across the years with each month as an observation time point [82]. We used the RISmed package in R to quantify citation counts of each sandfly in the PubMed database. Overall, we collected 12 morphological traits and 25 different ecological and biogeographical traits.

Following (Evans et al. 2017 [24], Han et al. 2019 [23], Fischhoff et al. 2021 [90], Han et al. 2015 [91]), variables with less than 10 percent coverage and a correlation factor greater than 0.7 with other variables were not included in the final analysis to avoid overfitting and misestimating the importance of highly correlated variables. The cutoff removes traits for which less than 10 percent of sandflies have data, to filter out variables with low coverage to simplify the models a bit. In theory, a data coverage cutoff is not necessary because of the way boosted regression treats missing data—it views the missingness as a 'common value' to group on. If one were to include all of the data regardless of coverage, low coverage variables that also have "low information" (either due to low coverage or due to the feature not being consequential for prediction) have low relative importance scores. As such, we used the 10% cutoff to simplify the model by removing variables that are likely to not contribute to model performance and predictions due to "low information". In order to avoid cyclic analysis, we did not include the trait for biting humans in our analysis, as vectors are already defined to be anthropophilic. Traits with skewed distributions were normalized to avoid skewing our analysis with outlying and potentially influential data while training our models [91]. We used one hot encoding, which converts each categorical variable to a new binary variable with either a 0 or 1, to transform categorical variables into binary variables (eg. genus, shape of maxillary tip, or structure of hypopharyngeal teeth) [21].

## Data analysis

We used extreme boosting through the XGBoost library in Python to fit a logistic classifier boosted regression tree model (BRT). Extreme gradient boosted regression is a machine learning algorithm that creates an ensemble of weak decision trees to form a stronger prediction model by iteratively learning from weak classifiers and adding them to a strong classifier (i.e., boosting). Gradient boosted regression is flexible in that it allows for non-linearity, both among features (i.e., interactions) and between features and predictions, collinearity between features, and non-random patterns of missing data [21,22,90,91]. XGBoost also allows the use of regularization parameters to prevent overfitting models to small, unbalanced datasets. XGBoost additionally handles unbalanced data well by weighting positive labels, an advantage when analyzing our data set with relatively few known vectors and sparse feature coverage.

We fit two predictive models for the general *Leishmania* genus as there was not enough data to accurately make separate models for each species of *Leishmania*. For the primary model, only the confirmed vectors were used as positive labels (0: not a confirmed vector, 1: confirmed vector). We include an additional analysis that includes both confirmed and potential vectors as positive labels (0: no evidence of the sandfly carrying a *Leishmania* parasite, 1: confirmed vector or observed to carry a *Leishmania* parasite) (S5 and S6 Tables, and S6, S7, S8, S9, and S10 Figs). As such, the secondary analysis indicates the probability a sandfly may be naturally infected with *Leishmania*, but is not necessarily infectious upon infection.

For training and tuning analysis, the data was stratified and split into 80% training and 20% testing sets such that each set had an equal proportion of positive labels. To tune the

hyperparameters for our XGBoost model, we used the hyperopt library in Python, which uses Bayesian optimization to find the best performing parameters for the model. We define a search space to include parameters dealing with regularization, depth, and learning rate of the regression trees, then run the optimization algorithm to find the best performing parameters. To ensure the model was generalizable, we used a 3-fold nested cross validation process for parameter tuning, where the training dataset was divided into three folds or subsets. In cross validation, for 3 iterations, a combination of two of those folds was used as the training set, while the remaining fold was used for validation to optimize parameter estimation. The nested cross-validation approach provides conservative estimates of model performance when analyzing small datasets [92]. The training results are averaged over the folds to get the performance score of the model. Due to data sparsity, we opted for 3-fold nested cross validation rather than 10-fold [21]. We used the 10 best performing parameter sets (minimum log-loss) in our BRT models.

Since results of boosted regression tree models are often dependent on test/train splits [24], we used the 10 best performing sets of parameters, and 10 random test train splits, to train 100 total models using the XGBoost library in Python. We evaluated model performance across the 100 model iterations using the aggregate median of the Area Under the Receiver Operator Curve (AUC). Each of the models was used to generate a predicted probability for the sandflies by applying the trained models to the whole dataset of sandflies and their traits. The predicted probability ranged from 0 to 1, and we used the aggregate median generated across the 100 models to assess potential vector status. We define variable importance to be the number of times a variable is selected for splitting a regression tree, weighted by the improvement to the model as a result of that node. Importance was evaluated on a scale of 0 to 1, with higher numbers signifying that the variable had a higher impact on model training, and all the individual importances summing to 1. For the variables we converted from categorical to binary, the relative importance of each binary trait was summed to represent the importance for the overall categorical variable.

Our secondary model, which was trained and fit identically to the primary model, used both confirmed and potential vectors as positive labels. More information on the secondary model can be found in S5 and S6 Tables, and S6, S7, S8, S9, and S10 Figs.

To determine whether we were identifying traits of vectors and not only traits of well-studied sandflies (i.e., our model was not biased by study effort), we ran a citation prediction model. Using the same hyperparameter tuning technique, a gradient boosted regressor model, and citation count as the target variable, we generated predictions of citations and a trait profile of well-studied species. We then compared the trait profiles between our vector model and citation count model to determine bias due to study effort. Next, to test whether our model was overfitting and fitting spurious correlations in the data, we performed target shuffling for 50 iterations (i.e., randomly shuffled the response variable, vector status, for 50 model iterations) and got the average performance score. Target shuffling is a way to test the statistical accuracy of a model, and avoid identifying false positives through false patterns in the data [93]. The model is considered overfit if it identifies shuffled labels with a greater accuracy than with a coin flip (i.e., AUC $\leq$ 0.5).

## Results

Our models used data on known vectors to identify which sandfly species are the most likely to carry and transmit *Leishmania* parasites causing ACL. We trained our ensemble of boosted regression tree (BRT) models on the trait profile of the confirmed vectors, and predicted which species might be potential vectors by leveraging trait similarities among species. The models achieved a high aggregate median out-of-sample AUC of 0.86 with a standard error of

**Table 1. The median predicted probability, standard deviation, and percentile for sandfly species of unknown vector status with greater than 0.5 predicted probability of being a vector.** The infection status column indicates whether the sandfly is a potential vector (has been observed carrying the parasite, but not confirmed as transmitting it to humans; 'potential'), or that the sandfly has not yet been found infected with *Leishmania* in the wild ('unknown').

| species | probability | std | percentile | potential/proven |
|---|---|---|---|---|
| Psychodopygus amazonensis | 0,868 | 0,122 | 0,969 | potential |
| Nyssomyia antunesi | 0,852 | 0,122 | 0,963 | potential |
| Psychodopygus claustrei | 0,838 | 0,109 | 0,957 | unknown |
| Psychodopygus guyanensis | 0,748 | 0,133 | 0,938 | unknown |
| Pintomyia (Pintomyia) pessoai | 0,739 | 0,199 | 0,936 | potential |
| Psathyromyia (Psathyromyia) bigeniculata | 0,725 | 0,15 | 0,93 | unknown |
| Trichophoromyia auraensis | 0,702 | 0,195 | 0,928 | potential |
| Psychodopygus chagasi | 0,592 | 0,205 | 0,922 | unknown |
| Trichophoromyia castanheirai | 0,584 | 0,213 | 0,92 | unknown |
| Sciopemyia sordellii | 0,578 | 0,186 | 0,918 | potential |
| Psathyromyia (Psathyromyia) lanei | 0,556 | 0,182 | 0,916 | unknown |
| Evandromyia (Evandromyia) infraspinosa | 0,541 | 0,207 | 0,914 | unknown |
| Warileya rotundipennis | 0,511 | 0,217 | 0,908 | unknown |

0.008 across the 100 model iterations (S1 Fig); therefore, on average, our models classified 86% of our observations correctly.

For each sandfly species, we generated an aggregate median predicted probability score of how likely it is to be a vector of ACL, and the percentile rank of that possibility. We consider sandfly species to be potential vectors if our models assigned them a predicted probability over 0.5 on a scale of 0 to 1, where 1 indicates that the species has a highest probability of being a vector. The models assigned 35 of 37 confirmed vectors with a median predicted probability above 0.5 and 13 of 475 sandfly species of unknown vector status with a median probability above 0.5 (Tables 1 and S2 and Fig 2). Two confirmed vectors (*Pintomyia youngi*, *Pintomyia ovallesi*) had lower probability scores, which could be due to sparsity of data for those species, thereby limiting our models from predicting species that otherwise might have been vectors. All of the 13 unknown sandflies predicted above a 0.5 probability were above the 90th percentile, and 8 of those 13 were potential vectors that have been observed to carry *Leishmania* parasites but have not yet been confirmed as vectors (Table 1). The full list of sandflies and their predicted probabilities, percentile rank, and status can be found in S1 Table, and a map of confirmed and predicted vector species occurrence points appears in Fig 3. For our secondary model, which was trained on both proven and potential vectors, we generated the same types of predictions, resulting in a mean AUC of 0.86, and many of the same top predicted sandflies (S4 Table and S6 and S7 Figs).

The top two unknown sandflies predicted by the models were *Psychodopygus amazonensis* (mean probability = 0.868), which has been observed carrying *L. naiffi* in the wild, and *Nyssomyia antunesi* (mean probability = 0.852), which has been observed carrying *L. lindenbergi* (Table 1). The model also predicted *Psychodopygus claustrei* and *Psychodopygus guyanensis*, both of which have not been observed to carry any species of *Leishmania*, and *Pintomyia pessoai*, which can carry *L. braziliensis*. As such, our model suggests that not only can these sandflies become infected with *Leishmania* spp. that cause ACL, but they can also transmit the parasites and may be important vectors transmitting *Leishmania* spp. to humans, as well as among reservoir hosts. Our secondary model, which predicted the probability that sandflies can be naturally infected with zoonotic *Leishmania*, predicted *Psychodopygus amazonensis* and *Nyssomyia antunesi* with probabilities above 0.93, along with assigning *Psychodopygus claustrei*, *Psychodopygus guyanensis*, and *Pintomyia pessoai* probability scores above 0.75 (S5 Table).

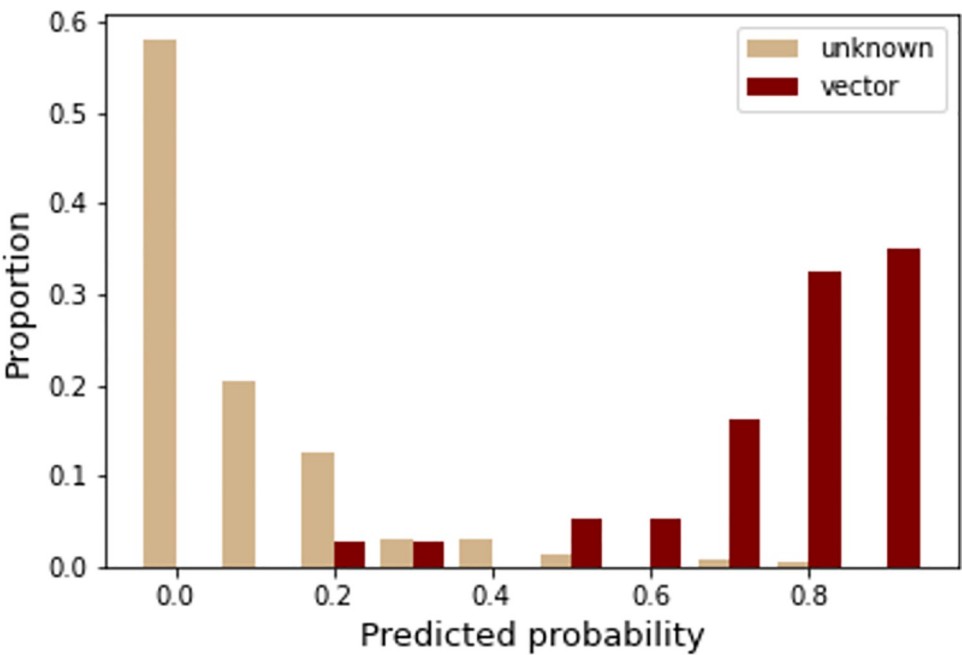

**Fig 2. The model accurately classifies known vectors and identifies relatively few species of unknown status as likely vectors.** A distribution of predicted probabilities of sandfly species separated by vector status, and scaled by percentage. Red bars indicate the proportion of confirmed vectors that were predicted at that probability, while beige bars indicate the proportion of sandfly species not previously identified as vectors that were predicted at that probability.

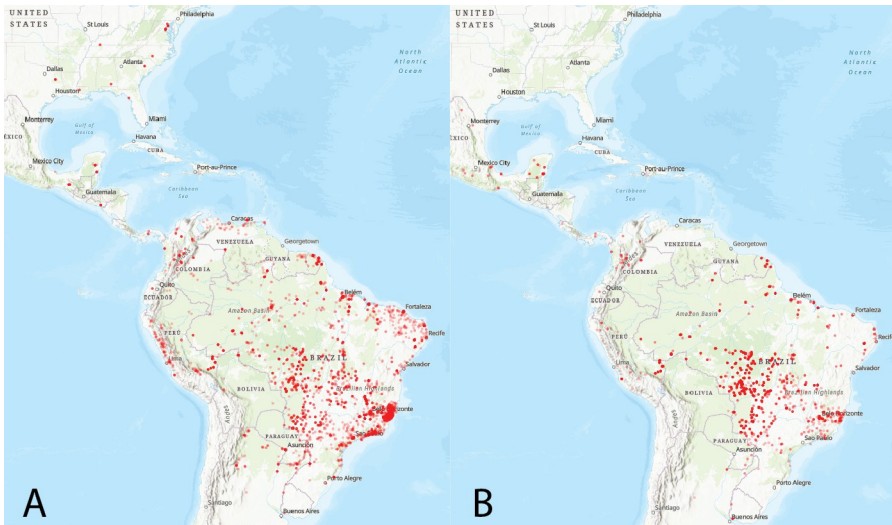

**Fig 3. Confirmed (A) and newly-predicted (B) vectors occur throughout the Americas.** (A) Observed occurrences of confirmed vectors of *Leishmania spp.* that cause ACL, taken from GBIF and plotted in ArcGIS (Esri, USGS | Esri, Garmin, FAO, NOAA, USGS) [78,79]. (B) Observed occurrences of sandflies of unknown vector status that our models assigned a predicted probability above 0.5. Most predicted vectors are in Brazil due to more extensive survey efforts and availability of public data [78,79]. Maps showing species richness and vector distribution for each species of *Leishmania spp.* can be found in S4 and S5 Figs.

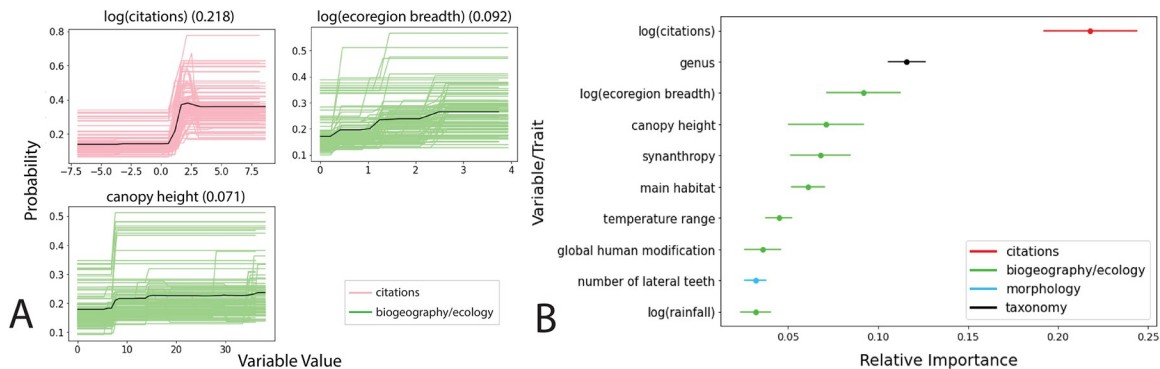

**Fig 4. Human biting, study effort, and canopy height were the most important features for predicting vector status.** (A) Partial dependence plots of the top three variables from the BRT analysis showing the marginal dependence of each trait (shown in order of importance) on the probability of being a vector of ACL. The variable along with its average importance (on a scale of 0–1) are above each plot, the trait value is shown on the x axis, and the effect on probability is shown on the y-axis. The colored lines represent the marginal dependence of the trait from the 100 BRT models, while the solid black line represents the average dependence. The definition of each variable can be found in S1 Table. (B) Variable importance, scaled from 0–1, for the top 10 most important variables with 95% confidence intervals. Points represent mean gain value across 100 iterations. The importances for binary variables were summed up to obtain a single value for the entire categorical variable.

The four most important features in our primary model (i.e., trained on confirmed vectors) are (i) the number of citations in PubMed, (ii) the genus of the sandfly, (iii) number of ecoregions the sandfly inhabits, and (iv) mean canopy height. Partial dependence plots (Fig 4B) indicate that sandflies that have greater study effort, live in areas with greater canopy height, and inhabit many different ecoregions are more likely to be *Leishmania* spp. vectors. Relative importance for the top twenty features and partial dependence plots for the top sixteen features are in S3 Table and S2 and S3 Figs. The trait profiles (partial dependence plots) of human modification index and forest land integrity index displayed opposite directionality, and our model assigned greater vector probabilities to sandflies that inhabit environments with less human modification and a greater land integrity index (S2 Fig). Other important variables included synanthropy (the tendency of an organism to live close to people and benefit from domestic habitats), main habitat, number of lateral teeth, along with environmental variables like rainfall, temperature, and temperature range, which are worth noting as climate drivers of sandfly development and survival [2,14,15]. The top features of our secondary model (i.e., trained on confirmed and potential vectors) were (i) the genus of the sandfly, (ii) the sandfly's main habitat, (iii) synanthropy, and (iv) number of ecoregions the sandfly inhabits (S6 Table).

Our target shuffling subanalysis returned a performance score of 0.50 (i.e., the model was no better at predicting shuffled labels than a coin-flip), indicating that our model is not overfit and simply finding spurious correlations in the data [93]. The citation prediction model was able to predict citation count to some extent ($R^2 = 0.18$), but has a different trait profile than that of our original vector models, suggesting that our predictions of *Leishmania* vectors do not simply reflect study bias (S3 and S4 Tables).

## Discussion

Our primary model leveraged ecological, behavioral, taxonomic, and biogeographic characteristics of sandfly species found across the Americas to predict the probability of a sandfly being a vector of ACL. The group of 100 BRT models was able to classify sandfly vectors with 86% accuracy and identified several previously unknown species with a relatively high probability of being a vector. Similarly, our secondary model was able to determine which sandflies might

be capable of carrying *Leishmania* with high accuracy. Overall, we found that the ecology and taxonomic features of a sandfly are most important in determining whether it has the potential to be a vector, followed by behavioral and morphological features. While citation count was the most important factor in predicting a vector, our citation count subanalysis suggested that our results were not primarily driven by study bias, as it performed poorly and had a different trait profile than our vector models. Study effort may indicate that *Leishmania* vectors are undersampled, but it may also indicate that vectors have a broad range that interfaces with human settlements, as it would be sampled in many different field surveys across the Americas. We found that both an increase in citation count and ecoregion breadth, i.e., the number of unique ecoregions in which the sandfly has been observed, were associated with a higher likelihood of a sandfly being a vector. This may suggest that the higher the species' propensity to adapt to and live in many different environments, the more likely it is able to survive—and be captured and sampled—in an environment with human inhabitants and transmit the *Leishmania* parasite.

Recent studies have shown that reservoir host relative abundance increases but overall mammal diversity decreases with human modification, while sandfly density increases with mammal diversity and decreases with human modification [94]. Our results support these findings as our model shows that sandflies that occupy areas with lower human modification and higher land integrity are more likely to be vectors. Since the opposite is true for the effect of human modification on reservoir host communities, this may indicate that the highest risk of *Leishmania* spillover lies at the interface between human modification and intact forests [95]. Indeed, synanthropic sandfly species that live in domestic habitats, but are associated with higher-integrity land cover, were more likely to be vectors. Interestingly, canopy height, which can be indicative of forest intactness, was one of the most important features in our model. An increase in canopy height suggests older forest and trees, as well as more space along the bark for sandflies to breed and live. Based on our model, sandflies found in these preserved areas with high forest integrity and low human modification are more likely to be vectors of *Leishmania*. Since sandflies in general are weak fliers and typically prefer to stay within 30 to 300 meters of their breeding and living environment [2,95], our analyses support previous hypotheses that transmission can happen when people enter previously undisturbed areas and come into contact with sandflies [2]. Since synanthropic sandflies are also more likely to be vectors due to their proximity to humans, this contact and transmission is thought to occur at interfaces between intact forests and domestic settlements, as previously suggested. Additionally, canopy height was correlated with sandfly wing width, which was removed for model training, yet indicates that our model accounts for sandfly biology and relevant environmental interactions. It is difficult, however, to track spillover that occurs at these interfaces as temporal and spatial differences in vector and host habitat make it challenging to observe the alignment of human, reservoir host, and sandfly dynamics conducive to wildlife-sandfly-human transmission [96]. Further, overall data sparsity of sandfly occurrence points and traits, as well as vector and host interactions, makes modeling ACL transmission even more complicated.

The second most important predictor of vector status was a sandfly's genus, indicating that sandflies from certain genera can be more or less inclined to be a vector. This suggests certain traits that occur in these parts of the sandfly phylogenetic tree are important for vector status, but full sandfly phylogenies were unavailable, so we were unable to quantitatively asses the effect of genetic distance. This highlights the importance of careful taxonomic work as well as high quality genetic data for as many species as possible.

We also found non-linear relationships between vector probability and climate variables. Vectors that occur in habitats with high temperatures and temperature ranges were less likely to be vectors, indicating that there is an optimal temperature range in which vector

transmission occurs. Sandflies in environments with higher rainfall are also more likely to be vectors, indicating an optimal range of precipitation that might also support high sandfly population abundance. This suggests there is a landscape of differential vector transmission success, which is worth investigating to determine risk score across different habitats.

The analysis helped to confirm that some sandfly species observed to carry *Leishmania* in the wild but have not yet been confirmed as capable of transmitting to humans (i.e., potential vectors) are likely to be infectious. These predicted vectors should be empirically tested to determine if they are indeed vectors, and using genomic blood meal analyses [97], determine what reservoir hosts they feed on. In particular, efforts should focus on *Psychodopygus amazonensis* and *Nyssomia antunesi*. *Psychodopygus amazonensis* has been observed to carry *L. naiffi*, and shares a genus with eleven sandflies that are proven vectors of *L. braziliensis* and *L. naiffi* [22], with the model assigning three new sandflies of genus *Psychodopygus* probability scores above 0.5. *Nyssomyia antunesi* is an anthropophilic sandfly observed to carry *L. lindenbergi*, and there is strong evidence that it is a vector of ACL [24,98]. It is additionally taxonomically related to *Nyssomyia whitmani* and *Nyssomyia umbratilis*, both vectors of *L. braziliensis* and *L. guyanensis* [22,99,100].

Our secondary model trained on identifying potential vectors supported the sandfly predictions generated by our primary model. Sandflies predicted by the primary model were ranked highly by our secondary model. However, not all potential vectors (sandflies that have been observed to carry but not transmit *Leishmania*) were predicted to be highly likely vectors to humans, indicating that the ability to carry the parasite, while important, may not be representative of the sandfly's ability to be a human vector of ACL. Rather, there are additional biological variables like forest integrity, canopy height, and temperature range that play important roles in *Leishmania* transmission to humans, as indicated in our analysis. Importantly, while these sandflies may not be vectors involved in human transmission, they may still be involved in transmitting the pathogen among reservoir hosts, maintaining a reservoir community, and warranting further research.

The majority of occurrence points of newly identified sandfly vector species were found in Brazil. This could be because Brazil has the most sampling studies done on sandflies, but incidence of ACL is also highest in Brazil according to Pan American Health Organization (PAHO) [6]. So while Brazil is generally better studied, there is also high vector species richness and abundance, specifically with predicted vectors concentrated in central Brazil in Mato Grosso and Mato Grosso Do Sul. Certain areas in Brazil are well sampled, but there are many regions that are neglected and ought to be the focus of new sampling efforts in order to identify new vectors. Future sandfly sampling efforts should be concentrated in these areas predicted by the model, in addition to poorly explored regions with not many sampling surveys or efforts, such as the Caatinga biome (northeastern Brazil) and the western Amazon. Then, to assess vector competence and incrimination according to the Killick-Kendrick criteria, entomological studies should be carried out in disease hotspots, followed by human disease notification and prevention efforts [97]. We additionally identified predicted vector occurrence points in Madre de Dios, Peru, another hotspot of ACL transmission, the eastern coast of French Guiana, and northwestern Colombia. Our analysis suggests new vectors (*Psy. amazonensis*, *N. antunesi*, *Psy. claustrei*, *Psy. guyanensis*, *P. pessoai*, *Pa. bigeniculata*, *T. auraensis*, *Psy. chagasi*, *T. castanheirai*, *S. sordelli*, *Pa. lanei*, *Ev. infrapsinosa*, *W. rotundipennis*) to be incorporated into surveillance efforts in these regions.

Due to data sparsity and low coverage, we were unable to generate *Leishmania* species-specific models, as some species had only one or two confirmed vectors. Instead, we opted for a genus-wide model that includes all *Leishmania* parasites that cause cutaneous leishmaniasis in the Americas. Although it is more informative and reliable to have more data for training the

model, we lose the *Leishmania* species-specific vector transmission information in this general model. Additional studies into the *Leishmania* parasites themselves as well as vectors that transmit them would increase the model accuracy for predicting specific *Leishmania* transmission cycles. In addition, some traits had to be removed from the final model due to low coverage (< 10%). These included activity throughout the day, lifespan, and which taxa they feed on, which would be valuable to future work in sandfly vector transmission of *Leishmania*. Data sparsity along with gaps in basic knowledge about ACL transmission contributed to model uncertainty. For instance, low trait coverage for some species could affect how they are predicted. While we are relatively confident in sandfly species predicted with high probability, additional data are required to reach similar predictive confidence for species that are currently predicted with low probability. We are more likely to have failed to identify an unknown and understudied species as a vector than an unknown but well-studied species. Additionally, although North American sandflies were included in the models, there were few occurrence points for these species, and they did not have high vector prediction scores. This is another limitation of our analysis considering the rise in North American cases of leishmaniasis and potential range shifts of sandfly vectors. Specifically, more data about species occurrence, behavior, and morphology are necessary to deepen our understanding of ACL vectors and spillover transmission in human populations.

While our model generally performed well, it assigned two known sandfly vectors a probability score lower than 0.5. One of these sandflies was *Pintomyia youngi*, which is a confirmed vector of *L. braziliensis*, and a potential vector of *L. amazonensis*. Due to difficulty in taxonomic identification procedures, *Pintomyia youngi* could have been misidentified as *Lutzomyia townsendi* [31,101,102], which means that covariate and/or vector status data might be assigned to the wrong species. This can lead to misrepresentation in the model (that already relies on sparse data), which might explain the low probability score assigned to *Pintomyia youngi*. A more stringent taxonomic identification criteria will help with *Leishmania* studies worldwide, as well as with modeling efforts. The other species with a low model-predicted probability was *Pintomyia ovallesi*, a lesser-studied vector of ACL in Central America [24,103,104]. It is possible that our model failed to identify them due to a lack of data and bias towards Brazilian sandfly vectors.

By understanding the environmental features that promote specific sandfly vector species, we can better (i) understand ACL transmission cycles as they impact human risk, (ii) understand the potential impacts of human modifications such as land use change on ACL transmission, and (iii) predict how ACL transmission cycles will respond to global change in the future. Based on the traits most predictive of vector status, we hypothesize that human risk peaks at the interface of human, vector, and host communities in intact forest areas with high canopies and relatively low temperature variance. Further analysis of epidemiological data and surveillance data on hosts, parasites, and vectors could be used to test this hypothesis. In addition, predicted but not yet confirmed vectors of ACL, i.e., *Psychodopygus amazonensis*, *Nyssomyia antunesi*, *Psychodopygus claustrei*, *Psychodopygus guyanensis*, and *Pintomyia pessoai*, should be empirically tested for competence in the laboratory. If confirmed, vector control methods should be expanded to account for the new vectors. Similarly, sampling and public health efforts should target central-west Brazil, where predicted vectors are concentrated, as well as lesser studied areas such as northeastern Brazil and the western Amazon. As ACL increases across the Americas, sandfly species that fit the trait profile of other confirmed vectors— including dwelling in forest with high canopy and high integrity, biting humans, and spanning many ecoregions—are important targets for *Leishmania* surveillance to better identify reservoir host transmission cycles and risk factors for human spillover and to cost-effectively target the most likely potential vectors.

## Supporting information

**S1 Table. Trait table.** All traits used in the sandfly vector model, along with definitions and data sources.
(XLSX)

**S2 Table. Predicted probabilities for the primary model for sandflies assigned a probability score greater than 0.5.**
(XLSX)

**S3 Table. Variable importance for the top 30 most important variables in the primary model, with categorical variables summed.**
(XLSX)

**S4 Table. Variable importance for the top 20 most important variables in the citation model, without categorical variables summed.** The trait profile is different compared to the primary model trait profile, ensuring that our primary model is not simply predicting which sandflies are well-studied.
(XLSX)

**S5 Table. Predicted probabilities for the secondary model, for sandflies that are not confirmed vectors that have been assigned a probability score above the 90th percentile.**
(XLSX)

**S6 Table. Variable importance for the top 30 most important variables in the secondary model, with categorical variables summed.**
(XLSX)

**S1 Fig. A histogram of AUC scores across all 100 BRT models for the primary model.** The average AUC score was 0.851, and the median AUC score was 0.863. An AUC = 1.0 means the model is perfectly able to distinguish between the sandflies that are vectors and those that are not.
(TIFF)

**S2 Fig. Primary model partial dependence plots showing the marginal effect of each trait (shown in order of importance) on the probability of being a vector of ACL.** The trait value is shown on the x-axis, and the importance is shown on the y-axis. Colored lines represent the marginal dependence of the trait from the 100 BRT models, while the solid black line represents the average dependence. The definition of each variable can be found in S1 Table.
(TIFF)

**S3 Fig. Variable importance of the primary model for the top 20 most important variables predicting sandfly vector status.** Points represent mean gain value across 100 iterations and error bars represent 95% bootstrapped confidence intervals. Categorial variables are not summed here; each variable is left as it's own.
(TIFF)

**S4 Fig. Occurrence points of predicted sandflies from the primary model, taken from GBIF and plotted in ArcGIS (Esri, USGS | Esri, Garmin, FAO, NOAA, USGS) [78,79], colored by species to indicate species richness.**
(TIFF)

**S5 Fig. Occurrence points for confirmed vectors of each *Leishmania* species causing ACL, colored by sandfly species to indicate species richness.** Points taken from GBIF and plotted in ArcGIS (Esri, USGS | Esri, Garmin, FAO, NOAA, USGS) [78,79]. *Leishmania* species not

mapped (e.g. *L. waltoni*, *L. lindenbergi*, *L. enrietti*) have no confirmed vectors.
(TIFF)

**S6 Fig. A histogram of AUC scores for the secondary set of 100 BRT models using both potential and confirmed sandfly vectors as positive labels.** The average AUC score was 0.867, and the median AUC score was 0.869.
(TIFF)

**S7 Fig. A distribution of predicted probabilities from the secondary model of sandflies separated by vector status, and scaled by percentage.** Red bars indicate the proportion of confirmed vectors that were predicted at that probability, while beige bars indicate the proportion of non-vector sandflies that were predicted at that probability.
(TIFF)

**S8 Fig. Variable importance for the top 10 most important variables in the secondary model with 95% confidence intervals.** Points represent mean gain value across 100 iterations. The importances for binary variables were summed up to obtain a single value for the entire categorical variable.
(TIFF)

**S9 Fig. Variable importance for the top 20 most important variables in the secondary model predicting sandfly vector status.** Points represent mean gain value across 100 iterations and error bars represent 95% bootstrapped confidence intervals. Categorial variables are not summed here; each variable is left as its own.
(TIFF)

**S10 Fig. Secondary model partial dependence plots showing the marginal effect (yhat) of each trait (shown in order of importance) on the probability of being a vector of ACL.** Variable value is shown on the x-axis, and marginal effect is shown on the y-axis. Partial dependence plots show the dependence of the probability on that trait's value, i.e., how the vector probability changes as the trait value increases.
(TIFF)

## Acknowledgments

The authors thank the Mordecai Lab at Stanford University for their feedback.

## Author Contributions

**Conceptualization:** Gowri M. Vadmal, Caroline K. Glidden.

**Data curation:** Gowri M. Vadmal, Caroline K. Glidden.

**Formal analysis:** Gowri M. Vadmal, Caroline K. Glidden.

**Funding acquisition:** Erin A. Mordecai.

**Investigation:** Gowri M. Vadmal, Caroline K. Glidden.

**Methodology:** Gowri M. Vadmal, Caroline K. Glidden, Barbara A. Han, Adrian A. Castellanos.

**Project administration:** Caroline K. Glidden, Erin A. Mordecai.

**Resources:** Gowri M. Vadmal, Caroline K. Glidden.

**Software:** Gowri M. Vadmal, Caroline K. Glidden.

**Supervision:** Caroline K. Glidden, Bruno M. Carvalho, Erin A. Mordecai.

**Validation:** Caroline K. Glidden, Bruno M. Carvalho.

**Visualization:** Gowri M. Vadmal, Caroline K. Glidden.

**Writing – original draft:** Gowri M. Vadmal, Caroline K. Glidden.

**Writing – review & editing:** Gowri M. Vadmal, Caroline K. Glidden, Barbara A. Han, Bruno M. Carvalho, Adrian A. Castellanos, Erin A. Mordecai.

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
