## [Decision Letter · Decision Letter 0]

13 Oct 2022

Dear Ms. Vadmal,

Thank you very much for submitting your manuscript "Data-Driven Predictions of Potential Leishmania Vectors in Latin America" for consideration at PLOS Neglected Tropical Diseases. As with all papers reviewed by the journal, your manuscript was reviewed by members of the editorial board and by independent reviewers. In light of the reviews (below this email), we would like to invite the resubmission of a significantly-revised version that takes into account the reviewers' comments. Please play close attention to the comments of reveiwer #2.

We cannot make any decision about publication until we have seen the revised manuscript and your response to the reviewers' comments. Your revised manuscript is also likely to be sent to reviewers for further evaluation.

Sincerely,

Mary Ann McDowell

Academic Editor

Alvaro Acosta-Serrano

Section Editor

Reviewer's Responses to Questions

**Key Review Criteria Required for Acceptance?**

**Methods**

-Are the objectives of the study clearly articulated with a clear testable hypothesis stated?

-Is the study design appropriate to address the stated objectives?

-Is the population clearly described and appropriate for the hypothesis being tested?

-Is the sample size sufficient to ensure adequate power to address the hypothesis being tested?

-Were correct statistical analysis used to support conclusions?

-Are there concerns about ethical or regulatory requirements being met?

Reviewer #1: The manuscript reports that the incidence of the increase in cases of Cutaneous Leishmaniasis is related to deforestation, climate change and globalization. I believe that there is a lack of understanding about the vectors and its parasites. Objective: Apply machine learning models (ecological niche models), to leverage the biological and geographic characteristics of vectors and potential vectors. In addition, generate profiles with confirmed vector characteristics and identify important factors for transmission. Very important ! The proposed methodology is adequate.

Reviewer #2: (No Response)

**Results**

-Does the analysis presented match the analysis plan?

-Are the results clearly and completely presented?

-Are the figures (Tables, Images) of sufficient quality for clarity?

Reviewer #1: The results give very relevantes data on vectors of Cutaneous Leishmaniais, related to corrected tanonomy.

Reviewer #2: (No Response)

**Conclusions**

-Are the conclusions supported by the data presented?

-Are the limitations of analysis clearly described?

-Do the authors discuss how these data can be helpful to advance our understanding of the topic under study?

-Is public health relevance addressed?

Reviewer #1: The resultas is in accordance with the proposal objetives and give more contribution to the knowledge to support suitable evidences on the impacts of climate changes on vetors of leishmaniases and its ecoepidemiology.

Reviewer #2: (No Response)

**Editorial and Data Presentation Modifications?**

Reviewer #1: xxxxxxxxxxxxxxxxxxxxxxxxxxxxxxxx

Reviewer #2: (No Response)

**Summary and General Comments**

Reviewer #1: This is a very interesting paper and must be published.

Reviewer #2: The authors here combine data compilation and machine learning to predict likely but currently unknown sandfly vectors of Leishmania. The work is generally sound, but I did have some major comments regarding (i) the geographic and taxonomic scope of the sandfly species considered (and broadening to the Americas as a whole), possible inclusion of other evolutionary features, and some details regarding the modeling methods itself. 

L14 and elsewhere: There are a few instances across the manuscript when parasites/diseases are conflated (e.g., “a parasitic disease transmitted by… on L14 and L34, “disease transmission” on L37, etc). The authors largely do a nice job referring to parasite transmission (given that parasites are transmitted between vectors and hosts, rather than the clinical diseases they cause), but a few additional cases could be resolved. 

L46 and L109: Clarify that the species being considered are exclusively in the subfamily Phlebotominae.

L53: The authors mention expansion of ALC cases into the southern US, but the models seem restricted to Central/South America. Are all sandfly species in North America also present in Central/South America? The taxonomic scale of the models is somewhat unclear. 

L109 and L125: Are the 512 sandfly species those restricted to the Americas as a whole or to only Central and South America?

L167: The authors mention genus here, but this is not mentioned in the earlier methods text on which traits were obtained. I also think the authors have missed an opportunity to also include evolutionary features of the sand fly species. Could the authors consider quantitative information (e.g., evolutionary isolation, phylogenetic diversity) from a sandfly phylogeny? Several other recent BRT-style papers have found such traits to be important. 

L161: A 10% feature coverage threshold seems rather low. Could the authors include a histogram of trait coverage across the 512 species? It would be helpful to see how coverage breaks down (e.g., if most of your features have very high coverage) and adjust your threshold to a higher cutoff (e.g., 50%), which should also improve your ability to predict novel vectors (if important traits have poor coverage). 

L189: When splitting data into training and test sets, did the authors perform stratified sampling via the positive labels? This would be important to ensure that the same fraction of positive labels are in both training and test splits, rather than one set having more proportional positives by random.

L199: Please include final parameter selections for the models. 

L232: If the authors are going to refer to the models as BRTs, I would use this earlier in the methods (e.g., within L171-179). 

L235: Can you provide not just the median but also a measure of variance (e.g., SE)?

L238 and L244-247: Because the authors use a 0.5 cutoff for predicted probability, the number of predicted-but-unknown vectors seems quite low. I would replace “some species” with “relatively few species” in the Figure 2 legend. The authors could keep the 50% cutoff here or consider instead using a 95% sensitivity threshold (e.g., Mull et al. 2022, Carlson et al. 2022). 

L268: One limitation of the maps here is that it is impossible to infer the density of different species. Could the authors consider also overlaying geographic distribution shapefiles of known/unknown vectors? Or coloring points by their corresponding vector species? The authors should also clarify from where these data are derived (e.g., GBIF?). Figure 3 also suggests the authors limit their data into northern Mexico, but I do wonder if expanding your dataset into sandflies present in North America would be interesting/worthwhile (especially giving some of the earlier framing in the Introduction about detecting range shifts of vectors with climate change).

L285 and L296: I think it’d be worth reminding the readers what differentiated the response variable in the primary and secondary models.

L319: Why not extent the species pool to those in North America (i.e., all Americas), especially given the earlier framing about climate shifts?

L320: I would suggest “identify several previously unknown species with a relatively high probability…”, to temper language here.

PLOS authors have the option to publish the peer review history of their article (what does this mean?). If published, this will include your full peer review and any attached files.

Reviewer #1: No

Reviewer #2: No
---

## [Decision Letter · Decision Letter 1]

24 Jan 2023

Dear Ms. Vadmal,

We are pleased to inform you that your manuscript 'Data-Driven Predictions of Potential Leishmania Vectors in the Americas' has been provisionally accepted for publication in PLOS Neglected Tropical Diseases.

Best regards,

Mary Ann McDowell

Academic Editor

Alvaro Acosta-Serrano

Section Editor

Reviewer's Responses to Questions

**Key Review Criteria Required for Acceptance?**

**Methods**

-Are the objectives of the study clearly articulated with a clear testable hypothesis stated?

-Is the study design appropriate to address the stated objectives?

-Is the population clearly described and appropriate for the hypothesis being tested?

-Is the sample size sufficient to ensure adequate power to address the hypothesis being tested?

-Were correct statistical analysis used to support conclusions?

-Are there concerns about ethical or regulatory requirements being met?

Reviewer #2: (No Response)

**Results**

-Does the analysis presented match the analysis plan?

-Are the results clearly and completely presented?

-Are the figures (Tables, Images) of sufficient quality for clarity?

Reviewer #2: (No Response)

**Conclusions**

-Are the conclusions supported by the data presented?

-Are the limitations of analysis clearly described?

-Do the authors discuss how these data can be helpful to advance our understanding of the topic under study?

-Is public health relevance addressed?

Reviewer #2: (No Response)

**Editorial and Data Presentation Modifications?**

Reviewer #2: (No Response)

**Summary and General Comments**

Reviewer #2: (No Response)

PLOS authors have the option to publish the peer review history of their article (what does this mean?). If published, this will include your full peer review and any attached files.

Reviewer #2: No

---

## [Editor Report · Acceptance letter]

13 Feb 2023

Dear Ms. Vadmal,

We are delighted to inform you that your manuscript, "Data-Driven Predictions of Potential Leishmania Vectors in the Americas," has been formally accepted for publication in PLOS Neglected Tropical Diseases.

Best regards,

Shaden Kamhawi

co-Editor-in-Chief

Paul Brindley

co-Editor-in-Chief
